# Long-Term Safety of Prenatal and Neonatal Exposure to Paracetamol: A Systematic Review

**DOI:** 10.3390/ijerph19042128

**Published:** 2022-02-14

**Authors:** Ram Patel, Katelyn Sushko, John van den Anker, Samira Samiee-Zafarghandy

**Affiliations:** 1Department of Medical Sciences, Schulich School of Medicine and Dentistry, Western University, London, ON N6A 5C1, Canada; rpate335@uwo.ca; 2Faculty of Health Sciences, School of Nursing, McMaster University, Hamilton, ON L8S 4K1, Canada; sushkokj@mcmaster.ca; 3Division of Clinical Pharmacology, Department of Pediatrics, Children’s National Health System, Washington, DC 20010, USA; JVandena@childrensnational.org; 4Division of Pediatric Pharmacology and Pharmacometrics, University of Basel Children’s Hospital, 4056 Basel, Switzerland; 5Division of Neonatology, Department of Pediatrics, McMaster Children’s Hospital, McMaster University, Hamilton, ON L8S 4K1, Canada

**Keywords:** paracetamol, acetaminophen, neurodevelopment, atopic disorders, reproductive disorders, neonatology, pharmacology

## Abstract

Introduction: Paracetamol is the most commonly used antipyretic and analgesic in pregnancy. It is also increasingly used off-label in the neonatal intensive care unit. Despite the frequent use of paracetamol, concerns have been raised regarding the high variability in neonatal dosing regimens and the long-term safety of early life exposure. Objective: To investigate the available evidence on the long-term safety of prenatal and neonatal paracetamol exposure. Methods: We conducted a systematic search of the electronic databases Ovid Medline, Ovid Embase and Web of Science from inception to August 2021 for original research studies of any design that described the use of paracetamol in the prenatal or neonatal (within the first four weeks of life) periods and examined the occurrence of neurodevelopmental, atopic or reproductive adverse outcomes at or beyond birth. Results: We identified 1313 unique articles and included 30 studies in the final review. Of all studies, 27 (90%), two (7%) and one (3%) were on the long-term safety of prenatal, neonatal and both prenatal and neonatal exposure, respectively. Thirteen (46%), 11 (39%) and four (15%) studies examined neurodevelopmental, atopic and reproductive outcomes. Eleven (100%), 11 (100%), and three (27%) studies on prenatal exposure reported adverse neurodevelopmental, atopic and reproductive outcomes. Only one study found a possible correlation between neonatal paracetamol exposure and long-term adverse outcomes. Conclusions: The available evidence, although limited, suggests a possible association between prenatal paracetamol exposure and an increased risk of neurodevelopmental, atopic and reproductive adverse outcomes. There is an immediate need for robust data on the long-term safety of paracetamol exposure in the prenatal and neonatal periods.

## 1. Introduction

Paracetamol, the most frequently used drug in pregnancy, is now one of the most common medications used off-label in the neonatal intensive care unit (NICU). Besides its antipyretic and analgesic properties, paracetamol promotes the closure of persistent patent ductus arteriosus (PDA) [1]. In 2010, paracetamol was the fifteenth most commonly used medication in NICUs in the United States. However, a recent study of European NICUs found that it is now the third most frequently administered drug, prescribed to over 65% of extremely preterm neonates [2].

The opioid-sparing effects of paracetamol and its comparable effectiveness to non-steroidal anti-inflammatory drugs (NSAIDs) in the closure of PDAs are the main drivers of its increasing use in the NICU [3,4,5]. Animal models have shown degenerative and necrotic changes in pulmonary, reproductive and nervous systems following exposure to paracetamol during the period of embryogenesis or organogenesis [6]. Over the past 20 years, growing human epidemiological data have also raised concern regarding early life exposure to paracetamol and an increased risk of neurodevelopmental, atopic and reproductive adverse outcomes [7]. Paracetamol’s analgesic and antipyretic effects are mediated through activation of descending serotonergic pathways, inhibition of prostaglandin synthesis and effects on cannabinoids receptors through its active metabolites [8,9,10]. The non-selective inhibition of peripheral cyclooxygenase and the interference with the cannabinoid receptors are among paracetamol’s mechanisms of action that could explain how such causal association could be biologically plausible [11]. We, therefore, aimed to conduct this review to systematically collate and assess the entirety of primary research on the long-term safety of prenatal and neonatal (within the first four weeks of life) exposure to paracetamol.

## 2. Materials and Methods

### 2.1. Protocol and Registration

The Joanna Briggs Institute Manual for Evidence Synthesis [12] and the Preferred Reporting Items for Systematic Reviews and Meta-Analyses (PRISMA) [13] guided the conduct and reporting of this review, respectively. The protocol for this review was submitted for registration to the international database of prospectively registered systematic reviews (PROSPERO, registration ID CRD42020212364) and was published [14].

### 2.2. Search Strategy

We developed a systematic search strategy with a professional librarian and searched the electronic databases Ovid Medline (1946 to August 2021), Ovid Embase (1974 to August 2021) and Web of Science (1900 to August 2021) (Table 1). We hand-searched the bibliographies of included studies for additional references. Using Google Scholar, we also searched for articles not commercially published, such as conference abstracts, dissertations and policy documents. We did not apply any language or design limitations.

### 2.3. Eligibility Criteria

We included original research studies of any design that described the use of paracetamol in the prenatal or neonatal (within the first four weeks of life) periods and examined the occurrence of neurodevelopmental, atopic or reproductive adverse outcomes. Studies were eligible for inclusion irrespective of the indication, dose, administration frequency and duration of paracetamol treatment. In the studies with a control group, we used the provided intervention(s), placebo or standard practice as the comparator. In studies with no comparator group, we collected the observational report of the long-term safety of paracetamol during the study period. Neurodevelopmental, atopic and reproductive adverse outcomes were any neurodevelopmental, atopic or reproductive disorders described by the authors of the primary studies, respectively. We excluded animal studies and duplicate studies.

### 2.4. Study Selection Process and Data Extraction

Following the database searches, using Covidence as the primary screening and data extraction tool, two reviewers (K.S. and R.P.) screened all articles independently at the title and abstract level. The same reviewers then assessed all full-text articles for eligibility. After identifying the studies to be included in the final review, the same reviewers extracted the data independently, using a prespecified standardized data extraction form [10]. We resolved any discrepancies in the study selection and data extraction processes through discussion with a third reviewer (S.S.Z).

### 2.5. Risk of Bias Assessment and Assessment of Evidence Certainty

Two reviewers (K.S and R.P) performed a qualitative assessment of the included studies, using standardized risk-of-bias assessment tools appropriate for each study design. These included the Newcastle–Ottawa Quality Assessment Scale for Cohort Studies [15], the modified tool for assessing the Quality of Modern Cross-Sectional Ecologic Studies [16,17,18] and the Cochrane Collaboration Risk of Bias 2 (RoB 2) tool for randomized controlled trials (RCTs) [19]. The same reviewers used the Grading of Recommendations Assessment, Development and Evaluation (GRADE) approach [20] to rate the certainty of the evidence, with adaptations by Murad et al. [21] to account for the absence of a single effect estimate. Any discrepancies were resolved through discussion with a third reviewer (S.S.Z.).

### 2.6. Synthesis Methods

We did not conduct a quantitative synthesis, due to the considerable heterogeneity of the included studies, methodologically, statistically and clinically. We performed a narrative synthesis of the study results based on the timing of exposure (prenatal or neonatal) and the primary outcome of interest (neurodevelopmental, atopic or reproductive adverse outcomes).

## 3. Results

We identified a total of 1552 articles. After removing duplicates, we screened 1313 studies at the title and abstract level and 97 at the full-text level. Title- and abstract-screening assessed the following: is the study about prenatal or neonatal exposure to paracetamol? Full-text screening assessed the following: does the study examine neurodevelopmental, atopic or reproductive outcomes? We included 30 studies in the final review (Figure 1) [22,23,24,25,26,27,28,29,30,31,32,33,34,35,36,37,38,39,40,41,42,43,44,45,46,47,48,49,50,51].

### 3.1. Characteristics of Included Studies

Of the 30 included studies, 13 (43%) were published within the last five years [24,25,26,27,29,31,34,36,37,45,47,50,51] and 21 (70%) within the last ten years [12,13,14,15,16,17,18,19,20,21,22,23,24,25,26,27,29,33,34,35,36,37,39,42,43,44,45,46,48,50,51]. Studies on the long-term safety of prenatal exposure to paracetamol comprised the majority of the included studies (n = 27, 90%) [23,24,25,26,27,28,29,30,31,32,33,34,35,36,37,38,39,40,41,42,43,44,45,46,47,48,49], with two studies reporting on the long-term safety of neonatal exposure to paracetamol (7%) [50,51] and one (3%) reporting on both timings of exposure [22]. Observational epidemiologic and cohort studies were the most common study designs (27, 90%) [23,24,25,26,27,28,29,30,31,32,33,34,35,36,37,38,39,40,41,42,43,44,45,46,47,48,49]. The study by Persky et al. was an RCT that randomized pregnant women to receive health education alone or health education plus education on reducing household asthma triggers. Education on paracetamol as a risk factor for the development of asthma was not discussed, and the authors collected maternal self-report of medication use in pregnancy and compared the occurrence of respiratory endpoints at one year of life among children of women who used and who did not use paracetamol during pregnancy [40]. As maternal paracetamol use was not randomized, we treated this study as an observational cohort for the purpose of our review. Two RCTs (7%) were included in the review [50,51]. The remaining study (3%) used an ecologic design [22] (Table 2 and Table 3; Appendix A).

### 3.2. Prenatal Exposure

For the studies that reported the long-term safety of prenatal paracetamol exposure, 13 (46%) [22,23,24,25,26,27,28,29,30,31,32,33,34], 11 (39%) [35,36,37,38,39,40,41,42,43,44,49] and four (15%) [45,46,47,48] focused on neurodevelopmental, atopic and reproductive adverse outcomes, respectively.

#### 3.2.1. Neurodevelopmental Adverse Outcomes

Of the 13 studies that examined the occurrence of neurodevelopmental adverse outcomes, 12 (92%) were observational cohort studies [23,24,25,26,27,28,29,30,31,32,33,34]. The other study (8%) used an ecologic design [22]. Of the 12 cohort studies, 10 used self-reported questionnaires [23,24,27,28,29,30,31,32,33,34] and two used umbilical cord and maternal plasma biomarkers [25,26] to assess prenatal exposure to paracetamol. Behavioral, performance, intelligence, executive and psychomotor function problems, autism spectrum disorder (ASD), attention deficit hyperactivity disorder (ADHD) and cerebral palsy (CP) were the neurodevelopmental outcomes assessed. Impaired neurodevelopment was reported in 11 studies, with specific reports of increased conduct problems (relative risk (RR) 1.35, 95% confidence interval (CI) 1.13–1.6), ADHD (odds ratio (OR) 1.88, 95% CI 1.18–3.00 and RR 1.13, 95% CI 1.01–1.27), hyperactivity symptoms (RR 1.22, 95% CI 1.04–1.43), motor milestone delays (OR 1.35, 95% CI 1.07–1.70) and hyperkinetic disorders (OR 1.37, 95% CI 1.19–1.59) among children with prenatal paracetamol exposure [23,24,25,26,27,28,29,30,32,33,34].

The ecologic study investigated the risk of prenatal paracetamol exposure and ASD using the national statistics on maternal use of paracetamol and the occurrence of ASD in nine countries. This study showed that each country’s average prenatal paracetamol use correlated with its ASD prevalence (r = 0.80, 95% CI 0.22–0.47) [22].

#### 3.2.2. Atopic Adverse Outcomes

Eleven studies investigated prenatal exposure to paracetamol and the risk of atopic adverse outcomes, using observational cohort (n = 10, 91%) [35,37,38,39,40,41,42,43,44,49] or epidemiologic (n = 1, 9%) [36] designs. The associations between paracetamol exposure and the occurrence of atopic adverse outcomes were identified via self-report (questionnaires, n = 8, 73% [36,37,38,40,41,42,43,49], interviews, n = 2, 18% [39,44]) and prescription records (n = 1, 9%) [35]. The study by Sordillo et al. also used serum levels of IgE as a marker of atopic adverse outcomes [43]. All studies (n = 11, 100%) reported an independent association between prenatal exposure to paracetamol and the risk of atopic adverse outcomes in childhood [35,36,37,38,39,40,41,42,43,44,49].

#### 3.2.3. Reproductive Adverse Outcomes

Four cohort studies investigated prenatal exposure to paracetamol and the risk of reproductive adverse outcomes [45,46,47,48]. All studies used maternal self-administered questionnaires to identify the trimester-based exposure to paracetamol. The occurrence of reproductive adverse outcomes was assessed via physical examination (n = 3, 75%) [46,47,48] and parent or child self-report (n = 1, 25%) [45]. Anogenital distance (AGD) (n = 2, 50%) [46,47], hypospadias or cryptorchidism (n = 1, 25%) [48] and the timing of pubertal development (n = 1, 25%) [45] were the target outcome variables. Three studies (75%) found a strong relationship between prenatal exposure to paracetamol and the development of reproductive adverse outcomes [45,46,48]. One of the two cohort studies that examined the association between prenatal analgesic exposure and AGD in offspring did not find a significant association between prenatal paracetamol exposure alone and the risk of shorter AGD at three months of age. However, the combined exposure to paracetamol and NSAIDs was significantly associated with reduced AGD at three months of age (AGD −4.1 mm; 95% CI 6.4–1.7) [47].

### 3.3. Neonatal Outcomes

Three studies examined long-term safety outcomes among neonates exposed to paracetamol [22,50,51]. Two studies (67%) focused on neurodevelopmental adverse outcomes only [22,51]. The other study (33%) examined neurodevelopmental and atopic adverse outcomes [51].

#### Neurodevelopmental and Atopic Adverse Outcomes

The first study used an ecologic design to investigate the country-level association between the rate of circumcision as an index of early life exposure to paracetamol and the rate of ASD. A strong correlation (r = 0.98) was found between circumcision and rate of autism for countries with data after 1995 [22]. The second study was a follow-up of an RCT that examined neurodevelopmental outcomes among neonates who received paracetamol or ibuprofen for PDA closure at 25 to 30 weeks’ gestation. There were no significant differences in neurodevelopmental impairment according to the Bayley Scales of Infant Development the Mental Developmental Index or the Psychomotor Developmental Index at 18 and 24 months of age [50]. The third study that focused on neurodevelopmental and atopic adverse outcomes was an RCT that examined paracetamol compared to a placebo for PDA closure in neonates 23 to 31 weeks’ gestation. There was no significant difference in neurodevelopmental or atopic outcomes between the two groups at two and five years of age [51].

### 3.4. Risk-of-Bias Assessment and Assessment of Evidence Certainty

#### 3.4.1. Risk-of-Bias Assessment

The risk of bias of the included observational epidemiologic and cohort studies [19,20,21,22,23,24,25,26,27,28,29,30,31,32,33,34,35,36,37,38,39,40,41,42,43,44,45] was assessed using the Newcastle–Ottawa Quality Assessment Scale for Cohort Studies [15]. All but three of the included observational studies received a rating between seven and nine, indicating a low risk of bias (Appendix A).

We used the RoB 2 tool [19] to determine the risk of bias of the included RCTs [46,47]. We rated the risk of bias of the included RCTs as low and with some concerns (Appendix A).

To assess the quality of the ecologic study by Bauer et al. [18], we used the modified tool for assessing the Quality of Modern Cross-Sectional Ecologic Studies [16,17,18] (Appendix A). The study received a rating of 17, indicating high study quality.

#### 3.4.2. Assessment of Evidence Certainty

We rated the certainty of the evidence on the long-term safety of prenatal paracetamol exposure as low. This was due to the observational design of the included studies and serious imprecision and inconsistency (Appendix A).

We rated the certainty of the evidence on the long-term safety of neonatal paracetamol exposure as moderate. This was due to very serious imprecision and serious inconsistency of the included studies (Appendix A).

## 4. Discussion

### 4.1. Summary of Findings

The results of the current systematic review suggest that prenatal exposure to paracetamol is associated with an increased risk of neurodevelopmental, atopic and reproductive adverse outcomes, independent of the timing and cumulative dosing of exposure. The ecologic study that examined the association between neonatal paracetamol exposure and neurodevelopmental adverse outcomes also demonstrated a correlation between neonatal exposure to paracetamol and neurodevelopmental adverse outcomes. However, neither of the two available RCTs on neonatal exposure to paracetamol showed an increased risk of neurodevelopmental or atopic adverse outcomes. Although two of the three available studies on neonatal exposure were RCTs, they were limited in their assessment of long-term safety outcomes, due to their small sample sizes. Furthermore, the exposure to paracetamol was limited to four days in duration and 200 mg/kg of cumulative dosing. In contrast, among the studies examining prenatal exposure, the duration of exposure and cumulative dosing were often more remarkable.

### 4.2. Comparison with Other Research

Studies on transplacental transfer of paracetamol have shown that paracetamol crosses the placental barrier rapidly via passive diffusion with significantly faster maternal-to-fetal than fetal-to-maternal transfer, transferring up to 50% of the original dose [52]. Besides the free cross through the placenta, maternal use of paracetamol has also been shown to be associated with placental CpG methylation [53]. Several studies have linked adverse placental growth and function and poor fetal and long-term health outcomes to placental DNA methylation [54,55,56]. Although paracetamol is mainly metabolized in the liver, evidence has shown that a small amount of para-aminophenol produced from paracetamol metabolism combines with arachidonic acid to produce *N*-arachidonoylaminophenol (AM404) [57]. AM404 is an indirect agonist at the cannabinoid receptors and has a disruptive effect on the normal function of the endocannabinoid system, an identified factor in the pathophysiology of neurodevelopmental disorders [8]. The endocannabinoid system has an important role in regulating early brain development through neuronal migration and proliferation of progenitor cells [58]. Exposure to paracetamol in the early stages of brain development could interfere with variable neuroprotein dynamics in different areas of the brain, such as the hippocampus and cerebral cortex, causing variable degrees of impairment in neurodevelopmental pathways and long-term neurodevelopment [9,59].

Paracetamol use is also associated with decreased glutathione levels, resulting in increased oxidant-induced inflammation and enhancement of T-helper type-2 pathways. This could be the biological basis for a link between paracetamol and atopic adverse outcomes [10]. The available evidence also points toward paracetamol causing a perturbance in the hormonal process, such as steroidogenesis and depletion of sulfated sex hormones [60], providing the biological plausibility for the potential alteration in the development of the fetal reproductive system that is associated with paracetamol exposure.

Paracetamol is a drug with multidirectional mechanisms of action consisting of inhibition of cyclooxygenases, involvement in the endocannabinoid system, serotonergic pathways, potassium and calcium channels and ultimately L-arginine in the nitric oxide synthesis pathway [9]. All of these mechanisms have key roles in hemostatic function of several regulatory systems. This raises remarkable controversy over the use of paracetamol during critical periods of organogenesis.

### 4.3. Strengths and Limitations

Our systematic review aimed to examine the available research on prenatal and neonatal paracetamol exposure and long-term safety outcomes. Although we collected a considerable amount of evidence on prenatal exposure, most studies used observational designs. Thus, these studies are vulnerable to the limitations of observational data, including the inability to establish cause–effect relationships. In addition, although two of the studies that focused on neonatal exposure were RCTs, the conclusions that could be drawn from their results were limited, due to the studies’ small sample sizes.

In our review protocol, we indicated that we intended to collect data on neonatal pharmacokinetics (PK) and pharmacodynamics (PD). We also planned to evaluate the short-term adverse events associated with neonatal paracetamol exposure [14]. As the included studies did not report on PK, PD or short-term adverse events, we were unable to synthesize data on these secondary outcomes.

## 5. Conclusions

The available evidence suggests a possible association between prenatal exposure to paracetamol and an increased risk of neurodevelopmental, atopic and reproductive adverse outcomes. This is supported by the current knowledge of paracetamol’s mechanism of action and biologic plausibility. However, the evidence is limited to observational studies and prone to biases, such as confounding by indication. Our review did not find an association between neonatal exposure to paracetamol and neurodevelopmental, atopic or reproductive adverse outcomes. However, the lack of quantity of the available evidence limits our conclusions. Considering the recent increase in early life exposure to paracetamol and the possible risk of long-term adverse outcomes, as shown in this review, there is an immediate need for robust data from using more rigorous methodologies. Until further data are available, the currently existing information needs to be investigated by the safety advisory committees of drug-regulatory agencies so that timely and appropriate labeling updates can be made and accessed by consumers and healthcare providers.

## Figures and Tables

**Figure 1 ijerph-19-02128-f001:**
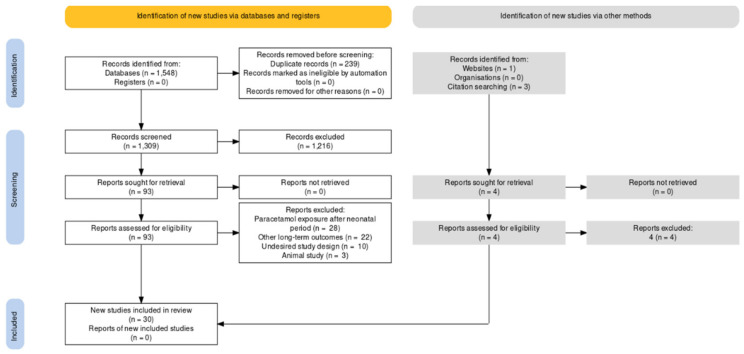
PRISMA flow diagram of the study’s selection process.

**Table 1 ijerph-19-02128-t001:** Ovid Medline, Ovid Embase and Web of Science search strategies.

Database: Ovid Medline
Exp Infant, Newborn/ Intensive Care Units, Neonatal/ Neonate *.mp Newborn *.mp Baby.mp Babies.mp Premie.mp Preemie.mp NICU.mp ((premature or preterm or pre-term or pre-mature or “small for gestational age” or postmature or term) adj infant or newborn).mp 1 or 2 or 3 or 4 or 5 or 6 or 7 or 8 or 9 or 10 Acetaminophen/ Paracetamol.mp Tylenol.mp 12 or 13 or 14 Pregnancy/ Prenatal Exposure Delayed Effects/ Abnormalities, Drug Induced/ ((pregnan* or prenatal or perinatal or antepartum or ante-partum or antenatal or ante-natal or fe?tus or fe?tal) adj exposure).mp 16 or 17 or 18 or 19 11 and 15 and 20
**Database: Ovid Embase**
Exp Newborn/ Neonatal Intensive Care Unit/ Neonate *.mp Newborn *.mp Baby.mp Babies.mp Premie.mp Preemie.mp NICU.mp ((premature or preterm or pre-term or pre-mature or “small for gestational age” or postmature or term) adj infant or newborn).mp 1 or 2 or 3 or 4 or 5 or 6 or 7 or 8 or 9 or 10 Acetaminophen.mp Paracetamol/ Tylenol.mp 12 or 13 or 14 Pregnancy/ Prenatal Exposure/ Drug Induced Malformation/ ((pregnan* or prenatal or perinatal or antepartum or ante-partum or antenatal or ante-natal or fe?tus or fe?tal) adj exposure).mp 16 or 17 or 18 or 19 11 and 15 and 20
**Database: Web of Science**
TS = (newborn * or neonat * or “baby” or “babies” or pre$mie * or “NICU”) TS = (“acetaminophen” or “paracetamol” or “tylenol”) TS = (pregnan * or “prenatal” or “perinatal” or ante$partum or ante$natal or fe$tus or fe$tal) #1 and #2 and #3

**Table 2 ijerph-19-02128-t002:** Characteristics of included studies on the long-term safety of prenatal paracetamol exposure.

Study	Design	Aim	Population	Sample Size	Age *	Exposure Time (WGA)	Dose (mg/kg) and Interval	Duration	Cumulative Dose	Outcomes	Assessment Time (Years)	Conclusion
					(years)				(mg/kg)			
Bauer 2013 [22]	Ecologic	Relationship of population weighted average of ASD and paracetamol use	20 studies on health outcomes and paracetamol use	N/R	N/R	N/R	Paracetamol	ASD	N/R	Biologic plausibility along with clinical evidence is linking paracetamol to ASD and abnormal NDV
N/R	N/R	N/R	N/R	N/R
Other
pharmacotherapy
N/R	N/R	N/R
Stergiakouli 2016 [23]	PC	Association between behavioral problems and paracetamol use	Cases	≤18:	29.1	≤18	Paracetamol	Behavior problems	7	Children with prenatal exposure to paracetamol have ↑ risk of behavior difficulties that are not explained bybehavior or social factors linked to paracetamol use
Mother–child pairs with paracetamol exposure	4415	−4.5	≤32
	≤32:		
	3381			N/R	N/R	N/R

Controls	4681	29.2	≤18	Other pharmacotherapy
Mother–child pairs without paracetamol exposure		−4.5	≤32	N/R	N/R	N/R
Andersen 2012 [35]	PC	Risk of asthma and paracetamol exposure	Cases	976	N/R	30 days before the 1st day of LMP -	Paracetamol	Asthma	Birth until the date of asthma diagnosis, death, emigration or the end of cohort follow-up	Robust association between maternal prenatal paracetamol exposure and the risk of asthma in offspring
		Mother–child pairs with paracetamol exposure			delivery			
					N/R	N/R	N/R	







		Controls	196,084			Other pharmacotherapy	
		Mother-child pairs without paracetamol exposure			


				N/R	N/R	N/R	

Bertoldi 2019 [24]	PC	Risk of NDV adverse outcomes with paracetamol exposure	Cases	Cases	Viva:	Viva:	Paracetamol	NDV outcomes	2-3	No strong evidence of a negative association between prenatal paracetamol use and cognition in early childhood
Cohorts of Viva and Pelotas studies: Mother–child pairs with exposure to paracetamol	Viva	32.5 (5.0)	T1/ T2,
	T1/ T2: 837		T1 and T2
	T1 and T2:	Pelatos:	Pelotas:
	487	27.1 (6.6)	T1/ T2,
	Pelotas:		T1 and T2
	T1/T2:		T1/T2/T3
	2198		T1, T2 and T3
	T1 and T2:		
	1274			N/R	N/R	N/R







Control	Control	Viva:		Other Pharmacotherapy
Cohorts of Viva and Pelotas studies: Mother–child pairs without exposure to paracetamol	Viva:	T1/ T2				
	T1/T2: 361	32 (5.2)				
	T1 and T2: 692	T1 and T2				
	Pelotas T1/T2: 1620	32 (5.1)				
	T1 and T2: 2544	Pelotas:				
	T1/T2/	T1/T2 26.7 (6.7)				
	T3:	T1 and T2				
	1348	26.9 (6.6)				
	T1, T2 and T3:	T1/T2/ T3				
	3044	26.8 (6.6)				
		T1, T2 and T3				
		27 (6.6)		N/R	N/R	N/R
Ji 2020 [26]	PC	Association between cord plasma biomarkers of paracetamol exposure and risk of ADHD and ASD	Cases	Cases	<20:		Paracetamol	ADHD, ASD and other DDs	0–21	Prenatal paracetamol exposure is associated with an increased risk of ADHD and ASD
			Mother–child pairs with maternal cord paracetamol burden in the T1 or T2	T1: 332	65 (11.85)					
				T2: 332	20–34: 728 (70.99)					
					≥ 35: 183 (17.15)					


							N/R	N/R	N/R			
			Controls	Controls			Other pharmacotherapy			
			Mother–child pairs with maternal cord paracetamol burden in the T3.	T3: 332					
							N/R	N/R	N/R			
Garcia-Marcos, 2009[36]	Epidemiologic	Paracetamol exposure and prevalence of wheezing, modified by maternal asthma	Cases	Mothers with asthma and exposure:	N/R	N/R	Paracetamol	Wheezing	4-5	The frequent usage of paracetamol during pregnancy is associated with the prevalence of wheezing in offspring during preschool years
Survey	Pairs of mothers with asthma and their children exposed to paracetamol	34
	(1) ≥1	Non-asthmatic mothers with exposure:
		in pregnancy	901
		(2) ≥1/month		N/R	≥1	N/R
		pregnancy		≥1/m

						Other pharmacotherapy
			Asthmatic mothers without exposure:			
		Controls	31			
		Pairs of mothers with asthma and their children never exposed to paracetamol	Non-asthmatic mothers without exposure:			
			775			

							N/R	N/R	N/R
Golding 2020	PC	Paracetamol exposure and childhood behavioral and cognitive outcomes	Cases	Cases	N/R	18–32	Paracetamol	Cognitive and behavior outcomes	0.5–17	Prenatal paracetamol exposure is associated with child attention and hyperactivity and conduct problems in boys
[27]			Mother–child pairs with paracetamol exposure	5279			
			N/R	N/R	N/R




			Other pharmacotherapy
	Controls	Controls	N/R	N/R	N/R
		Mother–child pairs without paracetamol exposure	6, 746
Ji 2018 [25]	PC	Association between maternal	Cases	Above median	<35:	N/R	Paracetamol	ADHD, ASD and other DDs	7	Maternal plasma biomarkers of paracetamol are
plasma	Mother–child pairs with maternal paracetamol biomarkers of	668	965	associated with ↑ risk of ADHD
biomarkers of paracetamol and ADHD in offspring	(1) above median	Below median	≥35:	
	(2) below median	630	215	
				N/R	N/R	N/R	




				Other pharmacotherapy	
				N/R	N/R	N/R	


	Controls	Controls		
	Mother–child pairs with maternal paracetamol biomarkers of	1062		
	“no detection”			
Bakkeheim	PC	Paracetamol exposure andallergic disease in school-aged children	Cases	T1: 31	N/R	T1 and T2	Paracetamol	Primary: current asthma,	10	Paracetamol exposure in pregnancy was associated with allergic rhinitis, but not with asthma or allergic sensitization
2011 [44]	Mother–child pairs with paracetamol exposure		allergic sensitization, allergic rhinitis
			T2/T3: 32	Secondary:
				history of asthma,
				N/R	N/R	N/R	current wheeze,
				FeNO > 16.7 (ppb),
				Mild-to-moderate bronchial hyper-responsiveness,
			Controls	972	Other pharmacotherapy	severe bronchial hyper-responsiveness
			Mother–child pairs without paracetamol exposure	
				N/R	N/R	N/R	

Piler 2018 [37]	PC	Association between paracetamol exposure and asthma in offspring.	Cases	170	Cases:	<20	Paracetamol	Asthma	3, 5, 7 and 11	The combination of prenatal and postnatal paracetamol exposure leads to higher risk of asthma development
	Mother–child	<19: 0.8%
	pairs with exposure to paracetamol	20–24: 1.2%
		25–30: 1.5%	N/R	N/R	N/R
		>31: 2.2%





	Controls	No paracetamol exposure: 3324	Controls:	Other pharmacotherapy
	Mother–child pairs	paracetamol and aspirin exposure: 97	<18.5	
	(1) without paracetamol exposure	aspirin exposure: 532	18.5–24.9:	
	(2) with exposure to paracetamol and aspirin		<19: 41.6%	
	(3) with exposure to aspirin		20–24: 35.1%	
			25–30: 25.0%	
			>31: 30.9%	
					Aspirin	Aspirin	Aspirin
							N/R	N/R	N/R
Shaheen2002 [49]	LC	Association between paracetamol use and risk of wheezing and eczema	Cases	9400	N/R	<20	Paracetamol	Wheezing, eczema	Wheezing: 3.9	Frequent use of paracetamol in late pregnancy may ↑ the risk of wheezing in the offspring
Mother–child pairs with paracetamol exposure		
		20–32	Eczema:
			2.5
			N/R	N/R	N/R	
	Controls		
	Mother–child pairs		Other pharmacotherapy	
	(1) without paracetamol exposure		N/R	N/R	N/R	
	(2) with prenatal aspirin exposure		











Ernst 2019 [45]	LC	Association between paracetamol exposure and timing of pubertal development	Cases	8606	Cases:	<12	Paracetamol	Pubertal development	11.5 and q 6 ms until full sexual maturation or 18	Prenatal paracetamol use may have long-term effects on female offspring pubertal development
Mother–child pairs with paracetamol exposure	30.6 (4.4)	13–24
		>25




		Controls		N/R	N/R	N/R
		30.7 (4.4)	
	Controls				Other pharmacotherapy
	Mother–child pairs without paracetamol exposure			
		7216		







				N/R	N/R	N/R
Jedrychowski 2011 [39]	PC	Association between low exposure to paracetamol and eczemain early childhood and interaction with exposure to pollutants	Cases	Cases	28.82 (3.39)	T1/ T2/T3	Paracetamol	Eczema	0.3–2 -> q 3 ms	Very small doses of paracetamol in pregnancy may affect the occurrence of eczema in early childhood, but only with co-exposure to high concentrations of fine particulate matter
	Mother–child pairs with paracetamol exposure in T1/T2/T3	T1: 40	
		T2: 40	2–5 -> q 6 ms
		T3: 73	


			N/R	N/R	T1:	
			200	
			(500–1500)	
			T2:	
				1600	
				(500–1100)	
				T3:	
				2200	
				(500–8000)	
				T1/T2/T3: 2600	
				(500–1600)	
			Controls	Controls			Other pharmacotherapy	
			Mother–child pairs without paracetamol exposure in T1 or T2 or T3	T1: 283			
				T2: 283			N/R	N/R	N/R	
				T3: 248			
Vlenterie2016 [28]	PSM	Association between	Cases	20,749	<25: 155	N/R	Paracetamol			Psychomotor behavior/temperament problems	1.5	Long-term prenatal exposure to paracetamol associated with ↑ risk of motor milestone delay and impaired communication
Cohort	NDV impairment with exposure to paracetamol	Mother–child pairs with paracetamol exposure	25–29:						
			594						
			30–34:						
			805						
			≥35: 345						









		Controls	30,451	<25: 3069							
		Mother–child pairs without paracetamol exposure	25–29:						
			10,183						
			20–34: 11,87						
			>35: 5329						



















							N/R	N/R	N/R			
							Other pharmacotherapy					




							N/R	N/R	N/R			
Fisher2016[46]	PC	Relationship between	Cases			<8		Male:	0, 0.25, 1, 1.5 and 2	Prenatal paracetamol exposure during 8–14 weeks of gestation associated with shorter AGD
paracetamol	Mother–male child pairs with paracetamol exposure					AGD, penile length, testicular descent distance, cryptorchidism
intake and male infant genital development				14-Aug	Paracetamol	Female: AGD
				>14		
		33.5	33.5				3	
							(0.5–360.0).	
					N/R	N/R		
	Controls					
	Mother–male child pairs without paracetamol exposure				Other pharmacotherapy	
		456	33.5		N/A	N/A	N/A	
Lind 2017[47]	PC	Association between analgesic exposure and AGD in offspring	Cases	Paracetamol exposure only:	30.9	T1/T2		AGD	0.25	The negative association between prenatal analgesic exposure and AGD in male offspring suggests disruption of androgen action
Mother–child pairs with paracetamol exposure	365	Paracetamol
		N/R	N/R	N/R

Controls	No analgesic exposure:	
Mother–child pairs with exposure to	617	
(1) analgesic	Paracetamol and other analgesic exposure:	Other pharmacotherapy
(2) paracetamol and other analgesic	34	
(3) other analgesics	Other analgesics only:	
	11	
		N/A	N/A	N/A
Persky 2008 [40]	Cohort **	Association between paracetamol exposure and wheezing and allergic symptoms	Cases	Early pregnancy:	25.8	T1:		Wheezing, wheezing/	0.1, 0.25, 0.5, 0.75 and 1	Paracetamol use in middle to late but not early pregnancy may be related to respiratory symptoms
Mother–child pairs with exposure to paracetamol	344	[15,16,17,18,19,20,21,22,23,24,25,26,27,28,29,30,31,32,33,34,35,36,37,38,39,40,41,42,43]	16–27	Paracetamol	coughing emergency department visits, hospitalization, asthma
(1) early pregnancy	Middle-to-late pregnancy:		T2: 28–36		
(2) middle to late pregnancy	342				
				N/R	N/R	N/R	
Controls					
Mother–child pairs with	Aspirin: 9			Other pharmacotherapy	
(1) exposure to aspirin	Ibuprofen: 11				
(2) exposure to ibuprofen					
				N/R	N/R	N/R	
Petersen2018 [34]	PC	Association between exposure to paracetamol, aspirin or ibuprofen and risk of CP in offspring	Cases		Total cohort	T1		Overall CP, unilateral spastic CP	1 to 6	Children with T2 paracetamol exposure have ↑ risk of unilateral spastic CP
Mother–child pairs with paracetamol exposure		≤24: 19,318		Paracetamol
		25–29:	T2	
		66,037		
	91,015	30–34:	T3	
		70,268		N/R	N/R	N/R
		≥35:	
		29,994	



Controls	Aspirin: 5746			
Mother–child pairs with				
(1) aspirin exposure	ibuprofen: 7358			Other pharmacotherapy
(2) ibuprofen exposure				
				N/R	N/R	N/R
Rebordosa2018[41]	PC	Association between paracetamol exposure and asthma/ wheezing	Cases	Cases	T1/T2/T3	18 month cohort:	Paracetamol	Asthma/	1.5 or 7	Prenatal paracetamol exposure has moderate associations with asthma
Mother–child pairs with paracetamol exposure	18 month cohort:		<24:	bronchitis; wheezing; hospitalization for asthma
	54,530		5525	.
	7 year cohort:		25–29:	
	9546		25,601	
			30–35:	
			25,076	
			≥36:	
			10,147	
			7 year cohort:	
			<24:	
			1059	
			25–29:	
			4732	
			30–35:	N/R	N/R	N/R	
			5041	
			>36:	
			1864	














Controls				Other pharmacotherapy	
Mother–child pairs without prenatal paracetamol exposure	Controls			N/R	N/R	N/R	
	18 month cohort:			
	30,129			
	7 year cohort:			
	5981			
Ystrom2017[29]	PC	Association between prenatal paracetamol exposure and ADHD in offspring	Cases	Cases:	N/R	6 months before pregnancy	Paracetamol	ADHD	3 years until diagnosis	Long-term prenatal paracetamol exposure is associated with ADHD
	Mother–child pairs with prenatal paracetamol exposure	52,707	
			0–4, 5–8, 9–12, 13–16, 17–20, 21–24, 25–28,	N/R	N/R	N/R
		> 29, >30






	Other pharmacotherapy
	Controls	Controls:		N/R	N/R	N/R
	Mother–child pairs without paracetamol	60,266	
	Exposure		
Liew 2016 [30]	PC	Association between paracetamol exposure and behavior problems/HKDs in offspring.	Cases	Total Cohort:	30.8	Trimesters 1, 2 and 3	Paracetamol	Behavior problems,	5, 7	Prenatal paracetamol exposure appears to be associated with increased risk of behavioral problems and HKDs
Mother–child pairs with paracetamol exposure	1491	−4.4	HKDs
			N/R	N/R	N/R	








Controls		30.8		
Other pharmacotherapy
Mother–child pairs without prenatal paracetamol exposure		−4.2	N/R	N/R	N/R	
Rifas-Shiman2020 [31]	PC	Association between paracetamol and ibuprofen exposure and executive function/behavior problems in offspring				9.9	Paracetamol	Executive function and behavior problems	1, 8	Prenatal exposure to paracetamol is associated with poorer executive function in children
Cases	Total Cohort:	32.2
Mother–child pairs with paracetamol exposure	1225	−5.2	
			27.9


				N/R	N/R	N/R



Controls				Other pharmacotherapy
Mother–child pairs				N/R	N/R	N/R
(1) without paracetamol exposure			
(2) with ibuprofen exposure			

Streissguth,1987 [32]	PC	Association between paracetamol/ aspirin exposure and IQ/attention in offspring	Cases	Total Cohort:	N/R	22	Paracetamol	N/R	4	Prenatal paracetamol exposure was not associated with child IQ or attention
Mother–child pairs with paracetamol exposure	1529
	Follow-up at 4 years: 421	N/R	N/R	N/R
Controls	
Mother–child pairs	
(1) without paracetamol exposure	
(2) with aspirin exposure	
		Other pharmacotherapy
		N/R	N/R	N/R
Thompson 2014 [33]	PC	Association between paracetamol use and ADHD in offspring	Cases	Cases	N/R	N/R	Paracetamol	ADHD	7 and 11	Prenatal paracetamol exposure increases risk of
Mother–child pairs with paracetamol exposure	(437) 49.8	N/R	N/R	N/R	ADHD-like behaviors





Controls	Anti-inflammatory drugs: 11				
Mother–child pairs	−1.3	Other pharmacotherapy	
(1) without paracetamol exposure	Aspirin: 46 (5.3)	
(2) with exposure to other drugs	Antacid: 151				
	−17.4				
	Antibiotics: 204				
	−23.5				
		N/R	N/R	N/R	
Magnus2016 [42]	PC	Association between paracetamol exposure and asthma in offspring	Cases	Assessed at	<25: 1432 (30.1)	≤18	Paracetamol	Asthma	3 and 7 years	Prenatal paracetamol exposure independently associated with asthma-related outcomes
Mother–child pairs with paracetamol exposure	3 years:	25–29: 4927 (27.8)	≤30
	53,169	30–34: 5876 (27.8)	<6 months postpartum
Controls	7 years:	≥35: 2606 (27.3)		N/R	N/R	N/R
Mother–child pairs without prenatal paracetamol exposure	25,394			Other pharmacotherapy

		<25:
1712 (36.0)
25–29: 6522 (36.8)
30–34: 7820 (37.0)
≥35: 3857 (40.4)	N/R	N/R	N/R
Sordillo2015[43]	PC	Association between antipyretic exposure and asthma in offspring	Cases	992	32.2	Pregnancy	Paracetamol	Wheeze, asthma,	3–5, 7–10	Adjustment for respiratory infections in early life substantially diminished associations between infant antipyretics and early childhood asthma
Mother–child pairs with paracetamol exposure	−5.2	allergen sensitization





			N/A	N/A	N/A	
Controls	Without paracetamol exposure: 430		
Mother–child pairs			
	(1) without paracetamol exposure	With ibuprofen exposure:		
(2) with ibuprofen exposure	247		Alternative pharmacotherapy			
			N/A	N/A	N/A			
Snijder,2011 [48]	PC	Association between analgesic exposure and cryptorchidism/hypospadias	Cases	2388	29.96	Periconception;	Paracetamol	Cryptorchidism, hypospadias	2.5	Prenatal exposure to paracetamol
Mother–child pairs with paracetamol exposure	−75	−5.24		increases risk of cryptorchidism in offspring
			<14 WGA;	
			14–22 WGA;		
			20–32 WGA	
				N/R	N/R	N/R	



Controls	NSAID:			
Mother–child pairs with	414 (13)			
(1) No prenatal analgesic exposure				
(2) NSAID exposure	Other analgesic: 382 (12)			
(3) other analgesics exposure				Other pharmacotherapy	
				N/R	N/R	N/R	
Shaheen 2010 [36]	LC	Association of paracetamol	Cases	Total cohort 13,988	27.2	<18–20	Paracetamol	Wheezing, asthma, eczema, hay fever;	7, 7.5 and 8.5	Maternal antioxidant gene polymorphisms may modify the relation between prenatal acetaminophen exposure and childhood asthma
exposure and antioxidant genotypes on childhood asthma	Mother–child pairs with paracetamol exposure	−5	20–32	lung function
				N/R	N/R	N/R	atopy




Controls	Other pharmacotherapy
Mother–child pairs without paracetamol exposure	N/R	N/R	N/R

* Maternal age. ** Treated as an observational cohort study. Note: Mean (SD), Median (IQR); ↑, increased; ↓, decreased; <, less than; >, greater than; ≤, less than or equal to; ≥, greater than or equal to; ADHD, attention deficit hyperactivity disorder; AGD, anogenital distance; ASD, autism spectrum disorder; CP, cerebral palsy; DD, developmental disorder; FeNO, fractional exhaled nitric oxide; HKD, hyperkinetic disorder; IQ, intelligence quotient; LC, longi-tudinal cohort; LMP, last menstrual period; N/R, not reported; PC, prospective cohort; PSM, propensity score matching; T1, trimester 1; T2, trimester 2; T3, trimester 3; WGA, weeks gestational age.

**Table 3 ijerph-19-02128-t003:** Characteristics of included studies on the long-term safety of neonatal paracetamol exposure.

Study	Design	Aim	Population	Sample Size	GA(weeks)	BW (kg)	PNA at Exposure	Dose (mg/kg) and Interval	Duration	Cumulative Dose (mg/kg)	Outcomes	Assessment Time (years)	Conclusion
Bauer, 2013[22]	Ecologic	Paracetamol use in early childhood and ASD	Cases 9 countries with data on circumcision and ASD after 1995	N/R	N/R	N/R	N/R	Paracetamol			ASD	N/R	Country- and state- level correlations between indicators of prenatal and neonatal paracetamol exposure and ASD
								N/R	N/R	N/R			
			Controls 12 countries with data on circumcision and ASD before 1995					Other pharmacotherapy					
								N/R	N/R	N/R			
Oncel 2017 [50]	RCT Follow-Up	The effects of paracetamol vs. ibuprofen on NDV outcomes	Cases Preterm infants with PDA on oral paracetamol	30	28 (1.7)	0.99 (0.21)	2-3	Paracetamol			NDV outcomes; hearing and vision	1.5–2	No evidence for significant difference in NDV outcomes for paracetamol vs. ibuprofen
								15, q 6 h	3	N/R			
			Controls Preterm infants with PDA on oral ibuprofen	31	27.6 (1.9)	0.98 (0.18)		Other pharmacotherapy					
								Ibuprofen: 10, 5, 5, q 24 h.	2	N/R			
Juujarvi, 2021 [51]	RCT Follow-Up	The effects of IV paracetamol and NDV outcome	Cases Preterm infants with PDA on IV paracetamol	23	23 + 5 to 31 + 6	1.22 (0.43)	1–4	Paracetamol			NDV	2	No long-term adverse reactions of IV paracetamol
								20, 7.5, q 6 h	4	126			
			Controls Preterm infants with PDA on IV placebo	21	23 + 5 to 31 + 6	1.12 (0.34)		Other pharmacotherapy Placebo or standard practice					
								0.45% saline	4	N/R			

ASD, autism spectrum disorder; BW, birthweight; GA, gestational age; GDMS, Griffiths mental development scales; IV, intravenous; Mean (SD); N/A, not applicable; NDV, neurodevelopmental; N/R, not reported; PDA, patent ductus arteriosus; PNA, postnatal age; RCT, randomized controlled trial; Vs, versus.

## Data Availability

Not applicable.

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
