# Peer review of "Long-Term Safety of Prenatal and Neonatal Exposure to Paracetamol: A Systematic Review"

_ijerph, 2022, doi:10.3390/ijerph19042128_

Round 1

Reviewer 1 Report

The protocol has been previously published in 2020: Samiee-Zafarghandy S, Sushko K, Van Den Anker JLong-term safety of prenatal and neonatal exposure to paracetamol: a protocol for a systematic reviewBMJ Paediatrics Open 2020;4:e000907. doi: 10.1136/bmjpo-2020-000907

https://bmjpaedsopen.bmj.com/content/4/1/e000907

The review was well written. The authors did not conduct a quantitative synthesis due to the considerable heterogeneity of the included studies, both methodologically, statistically and clinically. Thus, they performed a  narrative synthesis of the study results based on the timing of exposure (prenatal or neonatal) and the primary outcome of interest (neurodevelopmental, atopic or reproductive adverse outcomes).

I have some minor comments:

The authors defined long-term safety as the occurrence of neurodevelopmental, atopic or reproductive adverse outcomes assessed at or beyond birth.

Comments: This is not very clear to me because the neonates were included also in this review. What is the duration post birth that you are examining in the cohort study?

We developed a systematic search strategy with a professional librarian and searched 62 the electronic databases Ovid Medline (1946 to August 2021), Ovid Embase (1974 to Au-63 gust 2021) and Web of Science (1900 to August 2021) (Figure 1).

Comment: Such info was not presented in Figure 1

We hand-searched the 64 bibliographies of included studies for additional references. Using Google Scholar, we also searched for articles not commercially published, such as conference abstracts, dissertations, and policy documents.

Comment:This should be reported in PRISMA

The studies included in these two review should be screened through to see suitability: Efficacy and Safety of Paracetamol for Patent Ductus Arteriosus Closure in Preterm Infants: An Updated Systematic Review and Meta-Analysis

https://www.frontiersin.org/articles/10.3389/fped.2019.00568/full

Paracetamol (acetaminophen) for patent ductus arteriosus in preterm or low birth weight infants

https://www.cochranelibrary.com/cdsr/doi/10.1002/14651858.CD010061.pub3/abstract

Figure 1. Ovid Medline Search Strategy.

Comment: this should be a table rather than a figure/image

Figure 2. PRISMA Flow Diagram of the Study Selection Process

Comment: Suggest to change to PRISMA 2020 format http://prisma-statement.org/prismastatement/flowdiagram.aspx

Author Response

Reviewer 1

The review was well written. The authors did not conduct a quantitative synthesis due to the considerable heterogeneity of the included studies, both methodologically, statistically and clinically. Thus, they performed a narrative synthesis of the study results based on the timing of exposure (prenatal or neonatal) and the primary outcome of interest (neurodevelopmental, atopic or reproductive adverse outcomes).

Abstract

  1. The authors defined long-term safety as the occurrence of neurodevelopmental, atopic or reproductive adverse outcomes assessed at or beyond birth – This is not very clear to me because the neonates were included also in this review. What is the duration post birth that you are examining in the cohort study?

Thank you for this comment. We have made the following change to the Abstract and Introduction of our study so that this information is presented clearly.

Page 1, Line 20:

Abstract:

In this systematic review, our objective was to investigate the available evidence on the long-term safety of prenatal and neonatal paracetamol exposure.

Changed to:

We conducted a systematic search of the electronic databases Ovid Medline, Ovid Embase and Web of Science from inception to August 2021 for original research studies of any design that described the use of paracetamol in the prenatal or neonatal (within the first four weeks of life) periods and examined the occurrence of neurodevelopmental, atopic or reproductive adverse outcomes at or beyond birth.

Page 2, Line 58:

Introduction:

In this systematic review, our objective was to investigate the long-term safety of prenatal and neonatal exposure to paracetamol.

Changed to:

We, therefore, aimed to conduct this review to systematically collate and assess all the primary research on the long-term safety of prenatal and neonatal (within the first four weeks of life) exposure to paracetamol and enable the best possible approach in its use.

Materials and Methods

  1. We developed a systematic search strategy with a professional librarian and searched 62 the electronic databases Ovid Medline (1946 to August 2021), Ovid Embase (1974 to Au-63 gust 2021) and Web of Science (1900 to August 2021) (Figure 1). – Such info was not presented in Figure 1

Thank you for this comment. We have made the following change to the manuscript:

Page 2, Line 78 (Table 1):

We have added the search strategies for Ovid Embase and Web of Science to Table 1.

  1. Figure 1. Ovid Medline Search Strategy. ­­– Comment: this should be a table rather than a figure/image

Thank you for this comment. We have made the following change to the manuscript:

Page 2, Line 78

We have changed the title of Figure 1 to Table 1 and have provided it in a table format, rather than an image format.

Results

  1. We hand-searched the 64 bibliographies of included studies for additional references. Using Google Scholar, we also searched for articles not commercially published, such as conference abstracts, dissertations, and policy documents. – This should be reported in PRISMA

Thank you for this comment. We had initially grouped all of the 1,552 records that we identified together under the box “Results identified through database searching and other sources.”

We have now made the following change to the manuscript:

Page 5, Line 130 (PRISMA Flow Diagram…)

We have separated the studies that we identified via Databases (Medline, Embase and Web of Science) and the studies that we identified through Other Methods (citation searching and Google Scholar). Other Methods includes the studies identified through our hand searches of the bibliographies of included studies [Citation searching (n = 3)]and the records identified through Google Scholar [Websites (n = 1)]. There was not an option in the software that the reviewer directed us to (http://prisma-statement.org/prismastatement/flowdiagram.aspx) to edit the sources of Other Methods from the pre-defined Websites, Organisations and Citation Searching. Therefore, Websites represents the study we identified via Google Scholar.

  1. The studies included in these two review should be screened through to see suitability: Efficacy and Safety of Paracetamol for Patent Ductus Arteriosus Closure in Preterm Infants: An Updated Systematic Review and Meta-Analysis (https://www.frontiersin.org/articles/10.3389/fped.2019.00568/full) and Paracetamol (acetaminophen) for patent ductus arteriosus in preterm or low birth weight infants (https://www.cochranelibrary.com/cdsr/doi/10.1002/14651858.CD010061.pub3/abstract)

Thank you for this comment. As per the reviewer's recommendation, we screened all the studies cited in these two articles and we did not find any new studies meeting the inclusion criteria for our systematic review. Therefore, no changes were required. 

  1. Figure 2. PRISMA Flow Diagram of the Study Selection Process – Comment: Suggest to change to PRISMA 2020 format http://prisma-statement.org/prismastatement/flowdiagram.aspx

Thank you for this comment. We have made the following change to the manuscript:

Page 5, Line 130

We used the link provided by the reviewer to create a PRISMA Flow Diagram that is formatted according to PRISMA 2020.

Reviewer 2 Report

Dear Authors,

The present study evaluates evidence on the long-term safety of prenatal and neonatal paracetamol exposure. The research subject is interesting and brings scientific important data in the field, as it deals with a subject that is currently of great interest. Some changes of the manuscript should nevertheless be performed in order to improve its quality. Following specific changes should thus be performed:

 Major changes

Abstract: this section should follow the structure of the manuscript: introduction, materials and methods, results, discussions, conclusions. The name of the ones that guided the study are not necessary, it is enough to mention it in the materials and methods.

Introduction: this section should needs serious changes. Firstly, it should contain information regarding similar existing studies in literature and, in comparison, authors should emphasize the novelty and originality of their study. It is not clear what the present study brings in novelty. It is not enough to add these informations in Discussions. Introduction should contain the purposes of these studies and in the Discussions details about results should be presented and commented. Secondly, introduction of this study into the large context is also not at all clear. Please add further informations and justifications. The purpose of the study should also be clarified. It would also be interesting to offer more details on the mechanisms underlying the biological activities of paracetamol in this section. It may help next sections and, as these information are still not clear, they would help the reader better understand this molecule and may represent a link with your study and a good starting point.

Results:

  • Please state the exclusion criteria in order to arrive at the 97 full articles assessed and also specify how did you select the 30 you included in the qualitative synthesis. I find this number of references quite reduced for a systematic review.
  • Tables 1 and 2 extend very much the length of the article and make it very difficult to read. They should be moved to supplementary and comments on the informations they contain should be included in this section and in Discussions. You may try to offer more critical comments when discussion the informations in these tables.

Discussions: it is not clear whether the cited existing studies have the same purpose as the present one. Please clarify and compare in order to highlight the novelty and originality.

Conclusions: Please offer potential perspectives for your study. You may even make a novel chapter or section, or you can also mix them up with Conclusions.

Bibliography should follow the recommendation of the journal.

All these suggested changes should be performed in order to bring further improvements to the manuscript. 

Author Response

Reviewer 2

Dear Authors, the present study evaluates evidence on the long-term safety of prenatal and neonatal paracetamol exposure. The research subject is interesting and brings scientific important data in the field, as it deals with a subject that is currently of great interest. Some changes of the manuscript should nevertheless be performed in order to improve its quality. Following specific changes should thus be performed:

Major changes 

Abstract

  1. This section should follow the structure of the manuscript: introduction, materials and methods, results, discussions, conclusions. The name of the ones that guided the study are not necessary, it is enough to mention it in the materials and methods.

Thank you for this comment. We have now changed our abstract as per this recommendation.

Page 1, Line 16

Abstract: Paracetamol is the most commonly used antipyretic and analgesic in pregnancy. It is also increasingly used off-label in the neonatal intensive care unit. Despite the frequent use of paracetamol, concerns have been raised regarding the high variability in neonatal dosing regimens and the long-term safety of early-life exposure. In this systematic review, our objective was to investigate the available evidence on the long-term safety of prenatal and neonatal paracetamol exposure. The Joanna Briggs Institute Manual for Evidence Synthesis guided the conduct of this review. We searched the electronic databases Ovid Medline, Ovid Embase and Web of Science from inception to August 2021. We identified 1,313 unique articles and included 30 studies in the final review, following independent screening by two reviewers. The majority of the included studies were observational and examined the long-term safety of prenatal paracetamol use. The available evidence, although limited, suggests a possible association between prenatal paracetamol exposure and an increased risk of neurodevelopmental, atopic and reproductive adverse outcomes. We did not find a similar association between neonatal exposure to paracetamol and long-term safety outcomes. However, the lack of quantity of the available evidence on neonatal exposure limits our conclusions. The results of this review call for immediate attention regarding the need for robust data on the long-term safety of paracetamol exposure in the prenatal and neonatal periods.

Changed to:

Abstract: Introduction: Paracetamol is the most commonly used antipyretic and analgesic in pregnancy. It is also increasingly used off-label in the neonatal intensive care unit. Despite the frequent use of paracetamol, concerns have been raised regarding the high variability in neonatal dosing regimens and the long-term safety of early-life exposure. Objective: To investigate the available evidence on the long-term safety of prenatal and neonatal paracetamol exposure. Methods: We conducted a systematic search of the electronic databases Ovid Medline, Ovid Embase and Web of Science from inception to August 2021 for original research studies of any design that described the use of paracetamol in the prenatal or neonatal (within the first four weeks of life) periods and examined the occurrence of neurodevelopmental, atopic or reproductive adverse outcomes at or beyond birth. Results: We identified 1,313 unique articles and included 30 studies in the final review. Of all studies, 27 (90%), 2 (7%) and 1 (3%) were on the long-term safety of prenatal, neonatal and both prenatal and neonatal exposure, respectively. Thirteen (46%), 11 (39%) and 4 (15%) studies examined neurodevelopmental, atopic and reproductive outcomes. Eleven (100%), 11 (100%), and 3 (27%) studies on prenatal exposure reported adverse neurodevelopmental, atopic and reproductive outcomes. Only one study found a possible correlation between neonatal paracetamol exposure and long-term adverse outcomes. Conclusion: The available evidence, although limited, suggests a possible association between prenatal paracetamol exposure and an increased risk of neurodevelopmental, atopic and reproductive adverse outcomes. There is an immediate need for robust data on the long-term safety of paracetamol exposure in the prenatal and neonatal periods.

Introduction

This section should needs serious changes.

  1. Firstly, it should contain information regarding similar existing studies in literature and, in comparison, authors should emphasize the novelty and originality of their study. It is not clear what the present study brings in novelty. It is not enough to add these informations in Discussions. Introduction should contain the purposes of these studies and in the Discussions details about results should be presented and commented.

Secondly, introduction of this study into the large context is also not at all clear. Please add further informations and justifications. The purpose of the study should also be clarified. It would also be interesting to offer more details on the mechanisms underlying the biological activities of paracetamol in this section. It may help next sections and, as these information are still not clear, they would help the reader better understand this molecule and may represent a link with your study and a good starting point.

Thank you for this comment. We have revised the Introduction as follows:

Page 1, Line 39:

Paracetamol, the most frequently used drug in pregnancy, is now one of the most common medications, used off-label, in the neonatal intensive care unit (NICU). Besides its antipyretic and analgesic properties, paracetamol promotes the closure of persistent patent ductus arteriosus (PDA) [1]. In 2010, paracetamol was the fifteenth most commonly used medication in NICUs in the United States. However, a recent study of European NICUs found that it is now the third most frequently administered drug, prescribed to over 65% of extremely preterm neonates [2].

The opioid-sparing effects of paracetamol and its comparable effectiveness to non-steroidal anti-inflammatory drugs (NSAIDs) in the closure of PDAs are the main drivers of its increasing use in the NICU [3-5]. Animal models have shown degenerative and necrotic changes in pulmonary, reproductive and nervous systems following exposure to paracetamol during the period of embryogenesis or organogenesis [6]. Over the past 20 years, growing human epidemiological data have also raised concern regarding early-life exposure to paracetamol and an increased risk of neurodevelopmental, atopic and reproductive adverse outcomes [7]. Paracetamol’s analgesic and antipyretic effects are mediated through activation of descending serotonergic pathways, inhibition of prostaglandin synthesis and effects on cannabinoids receptors through its active metabolites [8-10]. The non-selective inhibition of peripheral cyclooxygenase and the interference with the cannabinoid receptors are among paracetamol’s mechanisms of action that could explain how such causal association could be biologically plausible [11]. We, therefore, aimed to conduct this review to systematically collate and assess all the primary research on the long-term safety of prenatal and neonatal (within the first four weeks of life) exposure to paracetamol and enable the best possible in its use.

Results:

  1. Please state the exclusion criteria in order to arrive at the 97 full articles assessed and also specify how did you select the 30 you included in the qualitative synthesis. I find this number of references quite reduced for a systematic review.

Thank you for this comment. We have made the following change to the manuscript:

Page 4, Line 119

We identified a total of 1,552 articles. After removing duplicates, we screened 1,313 studies at the title and abstract level and 97 at the full-text level. We included 30 studies in the final review.

Changed to:

We identified a total of 1,552 articles. After removing duplicates, we screened 1,313 studies at the title and abstract level and 97 at the full-text level. Title and abstract screening assessed the following: Is the study about prenatal or neonatal exposure to paracetamol? Full-text screening assessed the following: Does the study examine neurodevelopmental, atopic or reproductive outcomes? We included 30 studies in the final review.

  1. Tables 1 and 2 extend very much the length of the article and make it very difficult to read. They should be moved to supplementary and comments on the informations they contain should be included in this section and in Discussions. You may try to offer more critical comments when discussion the informations in these tables.

Thank you for this comment. We reduced some of the information in Tables 1 and 2 in order to shorten the length of the article and provided the full tables as supplementary files.

Discussions:

  1. It is not clear whether the cited existing studies have the same purpose as the present one. Please clarify and compare in order to highlight the novelty and originality.

Thank you for this comment. The current study is a systematic review of the literature on long-term safety (neurodevelopmental, atopic or reproductive adverse outcomes) of prenatal and neonatal exposure to paracetamol. Therefore, we included all studies that had an objective that aligned with the objective of our systematic review: 1. Prenatal or neonatal exposure to paracetamol or 2. Assessment of one of the long-term outcomes (neurodevelopmental, atopic or reproductive adverse outcomes). To further clarify this, we have made the following changes to our manuscript:

Page 2, Line 58:  

Introduction: 

In this systematic review, our objective was to investigate the long-term safety of prenatal and neonatal exposure to paracetamol.  

Changed to: 

We therefore, aimed to conduct this review to systematically collate and assess all the primary research related to long-term safety of prenatal and neonatal (within the first four weeks of life) exposure to paracetamol so that we can provide additional data for the best clinical practice.  

Page 4, Line 119

Results: 

We identified a total of 1,552 articles. After removing duplicates, we screened 1,313 studies at the title and abstract level and 97 at the full-text level. We included 30 studies in the final review.

Changed to:

We identified a total of 1,552 articles. After removing duplicates, we screened 1,313 studies at the title and abstract level and 97 at the full-text level. Title and abstract screening assessed the following: Is the study about prenatal or neonatal exposure to paracetamol? Full-text screening assessed the following: Does the study examine neurodevelopmental, atopic or reproductive outcomes?   

Conclusions:

  1. Please offer potential perspectives for your study. You may even make a novel chapter or section, or you can also mix them up with Conclusions.

Thank you for this comment. We have made have made the following change to the manuscript.

Page 31, Line 331:

Conclusions

Changed to:

The available evidence suggests a possible association between prenatal exposure to paracetamol and an increased risk of neurodevelopmental, atopic and reproductive adverse outcomes. This is supported by the current knowledge of paracetamol's mechanism of action and biologic plausibility. However, the evidence is limited to observational studies and prone to biases, such as confounding by indication. Our review did not find an association between neonatal exposure to paracetamol and neurodevelopmental, atopic or reproductive adverse outcomes. However, the lack of quantity of the available evidence limits our conclusions. 

Considering the recent increase in early-life exposure to paracetamol and the possible risk of long-term adverse outcomes, as shown in this review, there is an immediate need for robust data using more rigorous methodologies. Until further data is available, currently available information needs to be investigated by the safety advisory committees of drug regulatory agencies so that timely and appropriate labelling updates can be made and accessed by consumers and health care providers.

Bibliography:

  1. Bibliography should follow the recommendation of the journal.

Thank you for this comment. We have revised the Bibliography so that it follows the recommendations of the journal.

Round 2

Reviewer 2 Report

Dear Authors,

The present study evaluates evidence on the long-term safety of prenatal and neonatal paracetamol exposure. Authors performed most of the suggested changes after the first round of review. Following specific changes should still be performed:

 Major changes

Abstract: Please shorten to 200 words, as suggested in the Instructions for Authors. You do not need to announce that a certain section is going to be described.

Introduction: Information regarding similar existing studies in literature are still not found. Is this the only study of a kind or are there similar ones? It is also still not clear what the present study brings in novelty compared with similar studies. The purpose of the study should still be clarified.

Results: Please explain why you only included 30 studies in your study. At this number of studies, I don’t know if the article type “Systematic Review” still fits, as it is not quite extensive.

Discussions: it is still not clear whether the cited existing studies have the same purpose as the present one. Neither novelty and originality are clear.

I don’t think the reference style matches the recommendations of the journal.

All these suggested changes should be performed in order to bring further improvements to the manuscript.